# Effectiveness of Quaternary Ammonium in Reducing Microbial Load on Eggs

**DOI:** 10.3390/molecules26175259

**Published:** 2021-08-30

**Authors:** Hao Yuan Chan, Anis Shobirin Meor Hussin, Nurul Hawa Ahmad, Yaya Rukayadi, Abd-ElAziem Farouk

**Affiliations:** 1Faculty of Food Science and Technology, University Putra Malaysia, Serdang 43400, Malaysia; samuel.tfa@gmail.com (H.Y.C.); nurulhawa@upm.edu (N.H.A.); yaya_rukayadi@upm.edu.my (Y.R.); 2Halal Products Research Institutes, University Putra Malaysia, Serdang 43400, Malaysia; 3Department of Biotechnology, College of Science, Taif University, P.O. Box 11099, Taif 21944, Saudi Arabia; farouk@tu.edu.sa

**Keywords:** quaternary ammonium compound, microbial control, egg shell

## Abstract

Table eggs are an affordable yet nutritious protein source for humans. Unfortunately, eggs are a vector for bacteria that could cause foodborne illness. This study aimed to investigate the effectiveness of a quaternary ammonium compound (quat) sanitizer against aerobic mesophilic bacteria, yeast, and mold load on the eggshell surface of free-range and commercial farms and the post-treatment effect on microbial load during storage. Total aerobic mesophilic bacteria, yeast, and molds were enumerated using plate count techniques. The efficacy of the quaternary ammonium sanitizer (quat) was tested using two levels: full factorial with two replicates for corner points, factor A (maximum: 200 ppm, minimum: 100 ppm) and factor B (maximum: 15 min, minimum: 5 min). Quat sanitizer significantly (*p* < 0.05) reduced approximately 4 log10 CFU/cm^2^ of the aerobic mesophilic bacteria, 1.5 to 2.5 log10 CFU/cm^2^ of the mold population, and 1.5 to 2 log10 CFU/cm^2^ of the yeast population. However, there was no significant (*p* ≥ 0.05) response observed between individual factor levels (maximum and minimum), and two-way interaction terms were also not statistically significant (*p* ≥ 0.05). A low (<1 log10 CFU/cm^2^) aerobic mesophilic bacteria trend was observed when shell eggs were stored in a cold environment up to the production expiry date. No internal microbial load was observed; thus, it was postulated that washing with quat sanitizer discreetly (without physically damaging the eggshell) does not facilitate microbial penetration during storage at either room temperature or cold storage. Current study findings demonstrated that the quat sanitizer effectively reduced the microbial population on eggshells without promoting internal microbial growth.

## 1. Introduction

Table eggs are unequivocally the most affordable and nutritious protein source for humans [1]. In the United States alone, 7.92 billion eggs were produced in 2018, and this production number is twenty-four times the country’s population of the same year [2]. Whereas, in Malaysia, the Federation of Livestock Farmers’ Association of Malaysia reported that approximately 12.5 billion eggs were produced in the year 2017 with a rate of more than 6 million table eggs per day [3]. Such a substantial production and export figure is certainly in tandem with the risk of food poisoning. For example, food poisoning caused by eggs was reported in Kuala Terengganu in May 2020 [4].

Egg categorization is not only done by size but can also be categorized based on various housing systems, e.g., (i) conventional cage, (ii) barn, (iii) free-range, and (iv) organic [5]. In the conventional cage, hens are heavily loaded in a confined cage with limited movement space and lay eggs throughout their lifetime. However, a recent resurgence of attention to animal welfare and healthier food choices, predominantly in developed countries, has shifted interest towards free-range or organic shell eggs [6,7].

On the contrary, organic and free-range farming systems allow hens to access open-air space and improve physical activities. Nevertheless, this freedom leads to the exposure of various environmental stressors such as weather variations, predation, contact with wild birds, viral or parasitic infection, or contaminants [8,9]. Ferrante et al. [10] demonstrated that free-range and organic eggs laid on the soil/grass are more likely to be dirtier than those laid in barn and cage housing systems without contact with soil. On the contrary, Alvarez-Fernandez et al. [11] reported that free-range and organic egg aerobic bacteria contamination have no significant difference from the conventional cage housing systems. In another study, Jones et al. [12] reported that conflicting results had been reported in several research findings conducted in Europe regarding eggshell microbial contamination in different housing systems.

Numerous methods, either through thermal or nonthermal decontamination methods, have been studied. Conventional heat treatment, e.g., hot air or hot water pasteurization, is among the most promising methods [13,14]. Alongside, nonthermal microbial control, e.g., chlorine, ultraviolet light, electrolyzed oxidized water, quaternary ammonium compound (quat) sanitizer, *N*-halamine, and sodium hypochlorite are also modern approaches due to their efficacy, low costing, and simple mechanism [15,16,17,18]. In a study, Wang and Slavik [19] reported a mere 3.4 and 6.7% of *Salmonella enteritidis* penetration into eggs after being washed with quat and sodium hypochlorite. In another study, Lucore et al. [20] asserted that a 3-log reduction of external shell bacteria after washing with a commercial egg wash detergent containing potassium hydroxide and potassium hypochlorite. A handful of studies have been conducted to challenge the efficacy of quaternary ammonium (quat) sanitizer against the microbial load on eggshells. Nonetheless, these studies did not extend to understanding the effect of quat on eggshell integrity during storage [21,22]. Hence, the present study intends to evaluate the microbial load of free-range and commercial farm shell eggs, the effect of quat sanitizer on the microbial load of free-range and farm shell eggs, and the external and internal microbial load of post-treatment free-range and commercial farm shell eggs.

## 2. Materials and Methods

### 2.1. Sample Collection

Two different brands of free-range shell eggs (one dozen) and two different brands of commercial farm shell eggs (one dozen) were purchased from a supermarket located in Mid Valley Megamall, Kuala Lumpur, Malaysia, and the 99 Speedmart retail store, Kuala Lumpur, Malaysia, including GPS locations (Table 1). The procurement from the different locations was due to stock availability. Only eggs free from apparent defects, cracks, and the latest expiry date were selected at purchase. The purchased samples were kept on ice in a polystyrene foam box during transportation to the Food Microbiology Teaching Lab, Faculty of Food Science and Technology, University Putra Malaysia, Selangor. The samples were immediately stored in a chiller (4 °C) upon arrival to the laboratory.

### 2.2. Part 1: Eggshell Microbial Enumeration

#### 2.2.1. Pre-Enrichment

Whole free-range shell eggs (*n* = three) and commercial farm shell eggs (*n* = three) from each brand were randomly selected from the egg tray and transferred into a sterile stomacher bag with autoclaved peptone water (0.1%; Merck KGaA, Darmstadt, Germany). Shell egg rinsate was drained into a sterile beaker after gentle shaking for 5 min. The rinsate was then pre-enriched in an incubator for 4 h at 35 °C followed by a serial dilution up to 10^−8^. A similar procedure was repeated for four different brands of shell eggs (two free-range and two commercial farms).

#### 2.2.2. Microbial Enumeration

The microbial load analysis was carried out with a 100 µL aliquot from the three highest dilutions pipetted into plate count agar (Oxoid CM0325, Hampshire, UK) in duplicate and incubated at 37 °C for 24 h for a total aerobic mesophilic bacteria enumeration. The visible colonies formed were calculated as the logarithm of the colony-forming units per eggshell (Log10 CFU/eggshell). The enumeration procedure was carried out in triplicate. Similar procedures were repeated for *Salmonella* spp. enumeration on Xylose Lysine Deoxycholate Agar (XLD) (Oxoid CM0469, Hampshire, UK) and incubated at 37 °C for 48 h; Coliform’s enumeration was performed on desoxycholate agar (Oxoid CM0139, Hampshire, UK) and incubated at 28 °C for 48 h; yeasts and molds were performed on Potato Dextrose Agar (PDA) (Oxoid CM0139, Hampshire, UK) and incubated at 25 °C for 5 days. MINITAB (Minitab, LLC, State College, PA, USA) application software (version 17) was used to analyze the mean population and variance (one-way ANOVA) between each brand.

### 2.3. Part 2: Quaternary Ammonium Compound (Quat) Sanitizer Treatment

#### 2.3.1. Pre-Washed Microbial Enumeration

Free-range shell eggs (*n* = eight) and commercial farm shell eggs (*n* = eight) were randomly selected for quat sanitizer treatment. Both free-range and commercial farm shell eggs were selected from the same producer in the interest of minimizing variation. Before treatment, the initial microbial count was carried out by swabbing 1 cm^2^ of the egg surface at three different spots using a sterile cotton bud. The bud was broken down and immersed into 10 mL of peptone water (Merck KGaA, Darmstadt, Germany) and vortexed. Microbial load analysis was carried out for total aerobic mesophilic bacteria, *Salmonella* spp., coliforms, yeasts, and molds.

#### 2.3.2. Quat Sanitizer Preparation

Commercially available quat sanitizer (Quat Sanitizer Brand X, Sanawang, Malaysia) was prepared according to the manufacturer’s recommended concentration for sanitizing unrinsed food contact surfaces, i.e., 1% (*v*/*v*) and 2% (*v*/*v*). 1 mL and 2 mL of quat sanitizer (5–10% alkyl dimethyl benzyl ammonium chloride) were mixed with 1 L of distilled water (room temperature), respectively. The concentration of the quat sanitizer was verified using quat test paper (Micro Essential Laboratory, Kuala Lumpur, Malaysia) by matching it to the color scale. A 150 mL of 1% (*v*/*v*) and 2% (*v*/*v*) sanitizer were subsequently poured into eight sterile stomacher bags.

#### 2.3.3. Quat Sanitizer Treatment

A full (2^2^) factorial experiment design was conducted with concentration as factor A (independent variable) and washing time as factor B (independent variable) as illustrated in Table 2. The shell eggs from the aforementioned “pre-wash microbial enumeration” procedure was placed into a stomacher bag filled with 100 mL of quat sanitizer (Quat Sanitizer Brand X, Sanawang, Malaysia), respectively, and gently shaken and washed.

When the washing time was up, the quat sanitizer was drained, and each treated egg surface was swabbed 1 cm^2^ from three different spots using a sterile cotton bud. The bud was broken down and immersed into 10 mL peptone water (Merck KGaA, Darmstadt, Germany) and vortexed. Microbial load analysis was carried out for total aerobic mesophilic bacteria, *Salmonella* spp., coliforms, yeasts, and molds.

### 2.4. Part 3: Post-treatment Storage Microbial Study

Commercial farm and free-range shell eggs treated with 200 ppm quat sanitizer for 15 min were selected for the post-treatment internal and external microbial load study. The main intention of such a selection was to investigate the vulnerability of the internal egg content toward vigorous washing with the highest permitted quat sanitizer concentration on the different types of shell eggs. Enumeration of total aerobic mesophilic bacteria, *Salmonella* spp., coliform, yeasts, and molds on both internal and external shell eggs were carried out at zero storage day. Treated shell eggs were then kept in a perforated stomacher bag (with small holes made from disinfected scissors with 75% alcohol) at room temperature and chilled temperature (0–4 °C) for 15 days before the expiry date, and up until the expiry date (Table 3) for further microbial load analysis.

## 3. Result and Discussion

### 3.1. Part 1: Eggshell Microbial Load

#### Total Aerobic Mesophilic Bacteria

The recovery of microbial load from free-range and commercial farm eggshell surfaces is illustrated in Table 4. Both brand A and B free-range shell eggs’ total aerobic mesophilic bacteria count were found to be more significant (*p* < 0.05) than brand C and D commercial farm shell eggs in 1 to 3 log10 CFU/shell egg. The highest bacteria count was 10 log10 CFU/shell eggs recovered from brand A free-range shell eggs, whereas the lowest count was 7 log10 CFU/shell eggs recovered from brand D commercial farm shell eggs. Likewise, the aerobic mesophilic bacteria recovered from eggshells reported by several researchers fell within 5 to 8 log10 CFU/shell eggs [5,23,24]. The differences in the aerobic mesophilic bacteria range between individual researchers can be ascribed to the different egg’s laying environment, including microbial level in the air and dust from the cage and the cleanliness of the packing area [5,12]. The study of Parisi et al. [23] elaborated that those free-range eggs tend to have more prolonged contact with hens after oviposition. Therefore, shavings of feces from hens could be a source of contamination.

Plate count agar used in the current study for bacteria enumeration was non-selective and non-inhibitory, allowing a wide range of bacteria cells to be grown on the media. Hence, various colony phenotypes (dimension, color, shape, etc.) were observed on the agar plate, as shown in Figure 1. At least five different types of colony phenotypes were observed in the current study, as shown in Figure 1. Furthermore, each brand was observed with different colony phenotypes, and brand D was observed with two different colony phenotypes on the same agar. This indicates that there might be at least 5 different strains of aerobic mesophilic bacteria present on eggshells. In a study, Chaemsanit et al. [24] exanimated 16 eggs from markets and discovered 116 aerobic mesophilic bacteria strains that belonged to 15 different genera: *Staphylococcus* spp., *Micrococcus* spp., *Enterococcus* spp., *Streptococcus* spp., *Bacillus* spp., *Corynebacterium* spp., *Acinetobacter* spp., *Neisseria* spp., *Salmonella* spp., *Proteus* spp., *Citrobacter* spp., *Escherichia coli*, *Klebsiella* spp., *Enterobacter* spp. and *Serratia* spp., *Streptococcus* spp. and *Micrococcus* spp. were the predominant genus’ amongst the isolates.

### 3.2. Salmonella spp. and Coliforms

On the contrary, *Salmonella* spp. and coliforms were undetected after 4 h pre-enrichment, on neither the free-range nor commercial farm shell eggs surface. Thus, even though the *Salmonella* prevalence is habitually linked to poultry and poultry-related products (as discussed in the literature review), the result of the current study may provide a discrete perspective of *Salmonella* contamination. This finding is inconsonant with Loongyai et al. [25], who asserts no observation of any *Salmonella* present on eggshell in a study of *Salmonella* prevalence from different laying hen housing systems located in Thailand. Similarly, a study by Ong et al. [26], observed that merely 10 out of 320 eggs (3.1%) were *Salmonella* spp.-positive in a study of *Salmonella* prevalence on a Malaysia poultry farm. Alongside this, Jones et al. [27] claim that the coliforms level is merely 0.64 Log CFU/mL from a conventional cage laying system. Thus, the contamination was constantly lower than the free-range system, where eggs were either in contact directly with the floor (dirt and grass) or the nest box. The authors also emphasize that floor contact was the primary source of coliforms contamination of the eggshell. In addition, eggshell microflora is also closely associated with geographical areas [28]. Thus, the absence of coliforms and *Salmonella* in the current study may indicate that both free-range and commercial farm shell eggs have minimum contact with the dirty floor with good husbandry practice in place.

### 3.3. Yeasts and Molds

The yeasts and molds population recovered from both free-range and commercial farm shell eggs are shown in Table 4. The yeasts population for both free-range and commercial farm shell eggs ranged between 2.74 to 6.96 log10 CFU/shell egg. However, there was no systematic trend observed when compared to the total aerobic mesophilic bacteria. An identical finding was reported by Jones and Anderson [29], where no significant difference of the yeast and mold populations were found in conventional cage and free-range nest box. The yeasts population from brand A and B free-range shell eggs appeared to have no significant difference (*p* > 0.05) with brand C commercial farm shell eggs. In contrast, brand D’s yeast population was peculiarly low (*p* < 0.05) with merely 2.7 log10 CFU/shell egg. Brand A yeasts population (as with its aerobic bacteria count) had the highest yeasts contamination, i.e., 6.9 log10 CFU/shell egg amongst all brands.

The highest mold population of 7.2 log10 CFU/shell eggs had been recovered from brand B free-range shell eggs. The prevalence of mold on eggshells is one of the deleterious factors that can cause the penetration of bacteria into the egg content. Molds’ hyphae on eggshell surfaces facilitate the enlargement of shell pores after ovulation which ease the entry of bacteria into the content of the eggs [28]. Brand D commercial farm shell eggs had the lowest mold population with merely 0.51 log10 CFU/shell egg. On the other hand, the mold population of brand C commercial farm shell eggs was approximately 2 log10 CFU/shell egg, significantly (*p* < 0.05) higher than the brand A free-range shell eggs. Likewise, the mold population between free-range and commercial farm shell eggs did not portray any systematic trend.

In terms of strain varieties, more than 14 different yeast and mold colony phenotypes were identified in the current study, and the colonies’ appearance is shown in Figure 2. In a study, Musgrove et al. [30] identified 380 yeast genus or species from eggshells which included *Candida*, *Cryptococcus*, *Hansenula*, *Hyphopichia*, *Metschnikowia*, *Rhodotorula*, *Sporobolomyces*, and *Torulaspora*. Almost 85% of the isolates (321 out of 380) were identified as *Candida* spp. *Candida famata* was the most identified species (*n* = 120), followed by *Candida lusitaniae* (*n* = 38). The authors also claimed that those identified yeast strains were generally non-infectious; however, some may cause opportunistic infections in immunocompromised individuals. Rajmani et al. [31] successfully isolated 129 molds from eggshells, and the isolated species belonged to six genera: *Aspergillus*, *Penicillium*, *Fusarium*, *Mucor*, *Rhizopus*, and *Alternaria*. *Aspergillus* (38.5%) was found the be predominant, followed by *Rhizopus* (20.51%), *Mucor* (11.28%), *Penicillium* (9.23%), *Alternaria* (6.66%), and *Fusarium* (6.66%).

### 3.4. Part 2: Efficacy of Quat Sanitizer Free-range and Commercial Farm Shell Eggs Microbial Load Reduction

Given the fact that the quat efficacy (microbial reduction) in both free-range and commercial farm shell eggs resembled due to similar treatment (same factor levels), thus the microbial log reduction (response) from both free-range and commercial farm shell eggs was integrated for data analysis. Figure 3 illustrates each factor level’s response (after data integration of both free-range and commercial farm shell eggs). Figure 4, Figure 5 and Figure 6 illustrate the comparisons of log reduction between free-range and commercial farm shell eggs, whereas Figure 7 illustrates the microbial colonies before and after treatment. The efficacy of quat sanitizer, in summary, significantly reduced approximately 4 log10 CFU/cm^2^ of the aerobic mesophilic bacteria to an acceptable level i.e., less than 6 log10 CFU/cm^2^; significantly reduced approximately 1.5 to 2.5 log10 CFU/cm^2^ of molds population to an un-detected level; significantly reduced (except free-range shell eggs) approximately 1.5 to 2 log10 CFU/cm^2^ of yeasts population to an un-detected level. Neither *Salmonella* nor coliforms were observed during pre- and post-quat sanitizer treatment. This result signifies the sanitation of shell eggs using 100 to 200 ppm quat sanitizer prior to processing of any non-cook or non-baked food/desserts, e.g., raw eggs mayonnaise, custard, mousses, etc., for the foodservice industry, could minimize food poisoning probability caused by shell egg contamination. Concerning the antifungal efficacy, the current study finding concurred with Bernardi et al. [32], whereby quat sanitizer exhibited a significant impact on molds, but yeasts seemed less susceptible. The authors reported that 2 to 2.9 log10 of *Penicillium roqueforti*, *Penicillium commune* and *Aspergillus brasiliensis* were observed after washing with a 2.5% quat sanitizer. *Candida albicans* was least susceptible to the quat sanitizer, whereby only a 1 to 1.9 log10 reduction had been observed. In another study, Bundgaard-Nielsen and Nielsen [33] demonstrated that a 2% quat sanitizer successfully reduced 2 to 4 log10 CFU/mL of *Penicillium*, Cladosporium, and Scopulariopsis, *Aspergillus* and *Eurotium*, *Neosartorya*
*pseudofischeri*, and *Monascus ruber* with a few exception strains that reduced more than 4 log10 CFU/mL. The antimicrobial properties of quat sanitizer are contributed to by the 12–14 alkyls chain length [34].

### 3.5. Variable Interaction by Two-Way ANOVA

The responses from each factor level were analyzed through a two-way ANOVA analysis. The four-combination factor level treatment significantly (*p* < 0.05) reduced the aerobic mesophilic bacteria. However, there was no significant (*p* ≥ 0.05) response observed between individual factor levels (maximum and minimum), and the two-way interaction terms were also not statistically significant (*p* ≥ 0.05). In addition, the relationship between each factor and the response may not depend on the value of the other factor. From the model summary, R2 = 4.35% indicates that whenever a variation had been observed in the value of y, only 4.35% of it is due to the model (or due to change in x), and 95.65% is due to error or some unexplained factor. The regression equation in uncoded units is defined as TPC = 4.15 − 0.00270 Concentration − 0.018 Time + 0.000270 Concentration*Time. Figure 8 illustrates the contour plot of the total aerobic mesophilic bacteria log reduction versus time and concentration. Likewise, each factor level and two-way interaction term for yeasts and molds population, illustrated in Figure 9 and Figure 10, were not statistically significant (*p* ≥ 0.05). From the yeasts model summary, R2 = 16.73% indicates that whenever a variation had been observed in the value of y, only 16.73 of it is due to the model (or due to a change in x), and 83.27% is due to error or some unexplained factor. The regression equation in uncoded units is defined as Yeasts = −1.11 + 0.0187 Concentration + 0.203 Time − 0.00151 Concentration*Time. From the molds model summary, R2 = 40.6% indicates that whenever a variation had been observed in the value of y, only 40.6% is due to the model (or due to a change in x), and 59.4 % is due to error or some unexplained factor. The regression equation in uncoded units is defined as Molds = −1.82 + 0.0170 Concentration + 0.395 Time − 0.002185 Concentration*Time.

The two-way ANOVA analysis outcome postulates that a higher concentration of quat sanitizer and washing time might be needed to observe significant log reduction. In a study, Bernardi et al. [32] demonstrated that no differences in fungi reduction were observed when the quat sanitizer concentration doubled from 2.5% to 5%. In another study of fungi inhabitation, Tubajika [35] reported that 5 to 10 % of fungi inhabitation had been observed only with a 10-fold increment of quat sanitizer concentration i.e., 100 to 1000 ppm. Even though the treatment period is considered a crucial factor in decreasing microbial load and heat treatment, e.g., pasteurization, autoclave, sterilization, etc., might not be the same in nonthermal treatment. The current study’s finding shows no significant (*p* ≥ 0.05) response was observed between the time factor level (maximum and minimum) and the two-way interaction terms with quat concentration. This explained no maximum exposure period required for the quat sanitizer to exhibit its efficacy but with a minimum of one minute of exposure as recommended by the producer on the labeling. Mustapha and Liewen [36] reported a similar finding that quat sanitizer successfully reduced more than 4 log10 CFU/mL of *Listeria monocytogenes* on a stainless-steel surface with low one-minute exposure.

### 3.6. Negative Control

Distilled water and dishwashing soap were used as a negative control in the current study to compare the quat sanitizer efficacy. The use of water and household dishwashing soap or laundry detergent to reduce eggshell microbial load can be traced back to the end of the 1940s [37]. Figure 11 illustrates the comparison of mean microbial reduction between quat sanitizer and both negative controls. Treatment duration (washing time) for the control study was identical to quat sanitizer 2^2^ experimental design, i.e., 5 min and 15 min; however, no solution concentration differences were identified in the control study (only a single factor). The reduction trend of aerobic mesophilic bacteria was observed as being identical with the quat sanitizer treatment, i.e., bacteria was significantly (*p* < 0.05) reduced after being washed with distilled water and dishwashing soap. Nonetheless, the mean difference was observed as lesser compared with the quat sanitizer. Only approximately 2.6 log10 CFU/cm^2^ of aerobic mesophilic bacteria reduction had been observed after distilled water treatment, and only approximately 2 to 2.4 log10 CFU/cm^2^ of aerobic mesophilic bacteria had been observed after dishwashing soap treatment. This finding suggests that washing eggs with water and household dishwashing soap cannot reduce the microbial count to a safe limit i.e., 6 log10 CFU/shell egg. Yeasts and molds population, on the contrary, showed no observed significant reduction (*p* ≥ 0.05) after distilled water and dishwashing soap treatment.

### 3.7. Part 3: Post-Treatment Storage Study

#### 3.7.1. Eggshell Microbial Population

Free-range and commercial farm shell eggs, which were washed with 200 ppm quat sanitizer for 15 min, were selected for the post-treatment external and internal microbial load study during the storage period up to the producer expiry date. Figure 12 illustrates the comparison of the external microbial load (log10 CFU/cm^2^) of free-range shell eggs during storage at room and chill temperature after being washed with 200 ppm quat sanitizer for 15 min. No statistical difference (*p* ≥ 0.05) was observed, at room temperature storage, for aerobic mesophilic bacteria and yeasts population along the storage period from post-treatment until 15 days before the producer’s expiry date. Molds population, on the contrary, increased from non-detected levels to 2.1 log CFU/cm^2^ along the storage period, from post-treatment until 15 days before the producer expiry date. When shell eggs were stored until the producer expiry date, the aerobic mesophilic bacteria population was astonishingly absent, but the yeasts increment and molds decrement were not statistically significant. Similar unsystematic trends were observed in cold storage (0–4 °C) during the storage period. The incremental and decremental aerobic mesophilic bacteria population along the storage period from post-treatment up to the producer expiry date were not statistically significant (*p* ≥ 0.05). Yeasts population decreased to non-detected levels from 1.3 logs CFU/cm^2^ after storage until 15 days before the producer expiry date in cold conditions. However, the population increased back to 2.3 log10 CFU/cm^2^ after storage until the producer expiry date. However, such an increment was not statistically different (*p* ≥ 0.05) with the post-treatment yeasts population. The molds population remained undetected after cold storage up 15 days before the producer expiry date. Nevertheless, the molds populations had increased to 2.2 log10 CFU/cm^2^ when shell eggs were stored until the producer expiry date.

Similar irregular trends were observed for the commercial farm shell eggs, and Figure 13 illustrates the comparison of the external microbial load (log10 CFU/cm^2^) during storage at room and chill temperature after washing with 200 ppm quat sanitizer 15 min. No statistical difference (*p* ≥ 0.05) was observed for aerobic mesophilic bacteria along the storage period from post-treatment to the producer expiry date at room temperature. Yeasts and molds population, on the contrary, significantly increased from non-detected to 2.8 log10 CFU/cm^2^ and 2.4 logs CFU/cm^2^ 15 days before the producer expiry date, stored at room temperature. Further storage until the producer expiry date showed no statistically (*p* ≥ 0.05) significant increment for the yeasts population, but the molds population, on the contrary, had a statistically (*p* < 0.05) significant increment from 2.4 to 3.5 log10 CFU/cm^2^. In cold storage, aerobic mesophilic bacteria were observed as non-statistically (*p* ≥ 0.05) incremental along with the storage from post-treatment to 15 days before the producer expiry date. However, aerobic mesophilic bacteria were observed to decrease to un-detected from 2.7 log10 CFU/cm^2^ after storage up to the producer expiry date. No detection of any yeast population throughout the storage study at cold storage was made. Likewise, the molds population had not been detected after post-treatment and storage until 15 days before the producer expiry date. However, mild increments had been observed from non-detected to 0.8 log10 CFU/cm^2^ on the producer expiry date.

An irregular, extreme microbial trend observed from both the free-range and commercial farm shell eggs during the storage study might indicate complications from uncontrolled storage conditions. A similar low aerobic mesophilic bacteria trend was observed in commercial farm shell eggs stored in a cold environment up to the producer expiry date. Storage of shell eggs in cold storage may preserve its freshness and, more crucial, microbial control. An article on egg safety published by the US Food and Drug Administration [38] has educated the public to purchase shell eggs sold from a refrigerator or refrigerated case and store them promptly in a clean refrigerator at a temperature of 4 °C or below.

#### 3.7.2. Internal Microbial Population

No microbial load was observed in both the free-range and commercial farm shell eggs’ internal contents over the storage time at either room temperature or cold storage in the current study. Thus, the finding postulates that washing with the highest permissible quat concentration, i.e., 200 ppm for 15 min discreetly (without physically damaging the eggshell) for both free-range and commercial farm shell eggs, will not facilitate microbial penetration during storage at either room temperature or cold storage. This finding is inconsistent with a study by Wang and Slavik [19] who demonstrated that quat sanitizer treatments did not damage the eggshell surface and reduced bacterial penetration for eggs stored for up to 21 days. In another study, Jones et al. [39] reported that aerobic bacteria were less than 1 log10 CFU/mL in egg content throughout the 10-week storage study at 4 °C, and yeasts and molds population were less than 0.3 log10 CFU/mL. The authors also illustrated that no significantly different egg content contamination between washed and unwashed shell eggs and eggs spoilage greatly depends on eggshell permeabilities [39].

Whether to wash and sanitize shell eggs remains an intensive debate in several countries, e.g., the United States of America, Japan, British, the European Union, etc. In the United States of America, all USDA-graded eggs and the largest volume processors are mandated to follow the washing step with a sanitizing rinse at the processing plant to minimize microbial contamination, especially *Salmonella* spp. [40]. Similar washing and sanitizing processes are also practiced by Japan; however, this practice remains elusive in Britain and the European Union for Class A eggs sold in the open market [37]. The lawmakers in the European Union reckon that washed eggs will cause moisture loss, pose damage to the physical barriers, such as the cuticle, and promote trans-shell contamination, thereby increasing the food safety risk to consumers if subsequent drying and storage conditions are not appropriate. Therefore, only authorized egg-washing systems to operate following the national guides are permissible to market washed eggs [41]. To recapitulate, egg sanitizing could render positive external microbial control during storage. Internal eggs content could be well preserved with the condition that appropriate sanitizer and washing systems are imposed.

### 3.8. Part 4: Post-treatment Storage Negative Control

#### 3.8.1. Eggshell Microbial Population

Commercial farm shell eggs underwent 15 min of distilled water treatment, and dishwashing soap treatment (washing) was selected for negative external and internal microbial load control study. Figure 14 illustrates the comparison of microbial load (log10 CFU/cm^2^) during storage at room and chill temperature after being washed with distilled water. The external aerobic mesophilic bacteria population increased non-statistically (*p* ≥ 0.05) along the storage period at room temperature up to the producer expiry date. Yeasts population had been observed to increase significantly (*p* < 0.05), approximately 0.9 log10 CFU/cm^2^ after storage for 15 days before the producer expiry date, and the population continued to increase another 0.4 logs CFU/cm^2^ on the producer expiry date. Whereas for the molds population, an increase was observed from non-detected levels during post-treatment and 15 days before the producer expiry date to 2.2 log10 CFU/cm^2^ during producer expiry date. In cold storage, the aerobic mesophilic bacteria population dramatically decreased (*p* < 0.05) from 4.4 log10 CFU/cm^2^ to 0.8 log10 CFU/cm^2^ 15 days before the producer expiry date but rebounded (*p* < 0.05) to 3.3 log10 CFU/cm^2^ on the producer expiry date. Yeast populations decreased statistically (*p* < 0.05) from 3.2 log10 CFU/cm^2^ to 2.5 log10 CFU/cm^2^ 15 days before the producer expiry date and further decreased to 1.1 log10 CFU/cm^2^ on the producer expiry date. Molds populations had not been observed throughout the storage study until the producer expiry date, i.e., 0.8 log10 CFU/cm^2^.

Figure 15 illustrates the comparison of the microbial load (log10 CFU/cm^2^) during storage at room and chill temperature after being washed with dishwashing soap. The external aerobic mesophilic bacteria population increased non-statistically (*p* ≥ 0.05) along the storage period at room temperature up to the producer expiry date. The yeasts population significantly increased (*p* < 0.05) approximately 2.3 log10 CFU/cm^2^ after storage for 15 days before the producer expiry, and the population were decreased approximately 0.9 log10 CFU/cm^2^ on the producer expiry date. Whereas for the molds population, an increase from un-detected levels during post-treatment and 15 days before the producer expiry date to 2.2 log10 CFU/cm^2^ during the producer expiry date was observed. In cold storage, the aerobic mesophilic bacteria population decreased dramatically (*p* < 0.05) from 4.3 log10 CFU/cm^2^ to 1.1 log10 CFU/cm^2^ 15 days before the producer expiry date but rebounded (*p* < 0.05) to 3.5 log10 CFU/cm^2^ on the producer expiry date. Yeast populations decreased statistically (*p* < 0.05) from 2.9 log10 CFU/cm^2^ to 1.3 log10 CFU/cm^2^ 15 days before the producer expiry date and further decreased to 0.8 log10 CFU/cm^2^ on the producer expiry date. The molds populations had not been observed throughout the storage study until on the producer expiry date, i.e., 1.2 log10 CFU/cm^2^. The fluctuation of the microbial count at both room temperature and cold storage might be due to the storage environment’s complication.

#### 3.8.2. Internal Microbial Population

No internal microbial load had been observed in both the distilled water treatment and dishwashing soap over the storage time at either room temperature or cold storage in the current study.

## 4. Conclusions

This study successfully demonstrated that two commercially available brands of free-range shell eggs’ aerobic mesophilic bacteria count were 1 to 2 log10 CFU/shell egg higher than two brands of commercial farm shell eggs. In addition, quat sanitizer had efficaciously reduced 4 log10 CFU/cm^2^ of the aerobic mesophilic bacteria; efficaciously reduced 1.5 to 2.5 log10 CFU/cm^2^ of the molds population to an un-detected level; efficaciously reduced (except free-range shell eggs) 1.5 to 2 log10 CFU/cm^2^ of the yeasts population to an un-detected level. In addition, cold storage microbial load was observed as lower than room temperature storage along the storage period.

The current study indicates that washing and sanitizing shell eggs purchased from the open market is vital in minimizing food poisoning incidents due to shell eggs contamination, especially in preparing non-cooked/baked foods. Quat sanitizer with a maximum concentration of 200 ppm is an effective alternative to minimize the microbial load yet is safe for non-rinse food contact. Purchased shell eggs should be stored in a cold environment before ensuring microbial loads are under a safe level after sanitizing with quat sanitizer.

Nonetheless, the sampling size could be increased in future studies to cover most of the commercial brands available in Malaysia’s open market. The impact on eggshell integrity after washing with quat sanitizer was demonstrated in the current study. However, a future improvement could be made by viewing the cross-section of the cuticle layer, the changes of egg yolk membrane integrity, the concentration of lysozyme and ovotransferrin over time due to washing, and prolonged storage. Also, comparison of various disinfectants e.g., hypochlorite, benzalkonium chloride, cetylpyridinium chloride, chlorhexidine, octenidine dihydrochloride, etc., could unravel the efficacy of microbial control of different sanitizers.

## Figures and Tables

**Figure 1 molecules-26-05259-f001:**
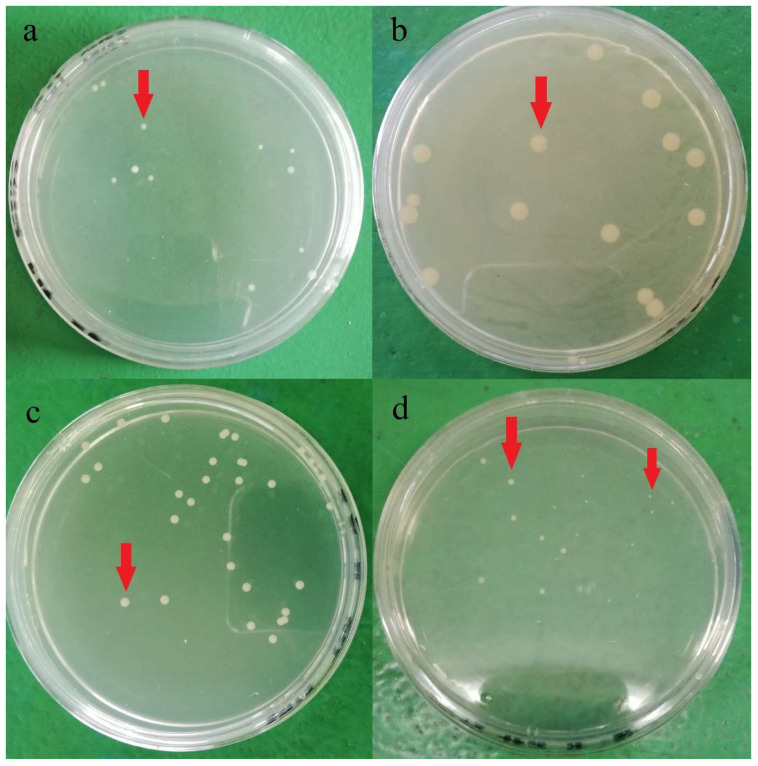
Different aerobic mesophilic bacteria colonies (red arrow) observed on plate count agar after 24 h incubation at 37 °C from a different brands of shell eggs: (**a**) brand A, (**b**) brand B, (**c**) Brand C, (**d**) Brand D.

**Figure 2 molecules-26-05259-f002:**
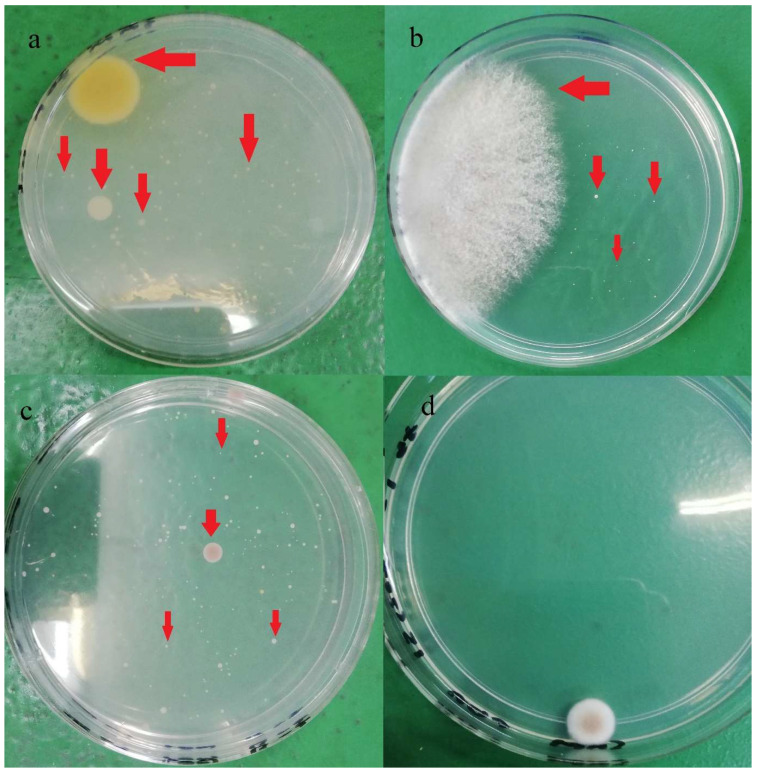
Different yeasts and molds colonies (red arrow) observed on PDA after five days incubation at 28 °C: (**a**) brand A, (**b**) brand B, (**c**) Brand C, (**d**) Brand D.

**Figure 3 molecules-26-05259-f003:**
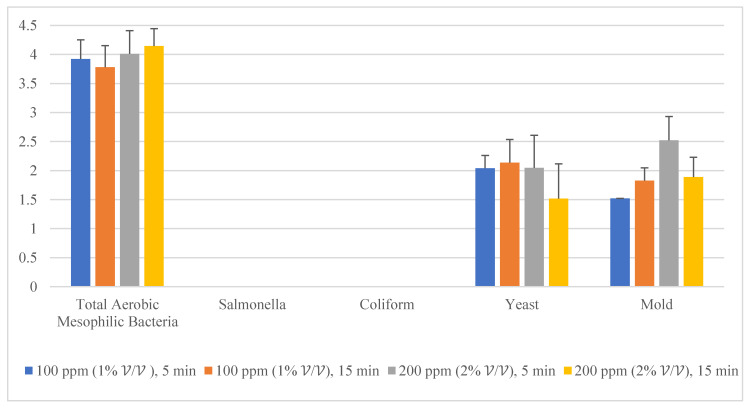
Total reduction of microbial load (log10 CFU/cm^2^) after quat sanitizer treatment.

**Figure 4 molecules-26-05259-f004:**
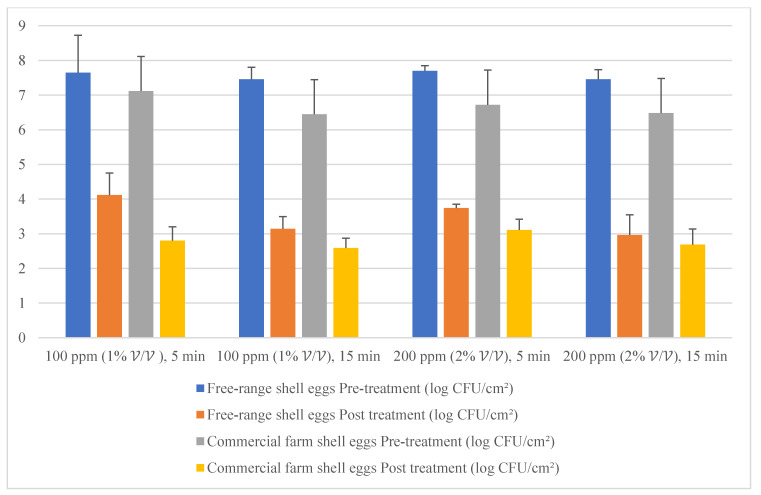
Mean population of aerobic mesophilic bacteria (pre- and post-treatment) for free-range shell eggs and commercial farm shell eggs.

**Figure 5 molecules-26-05259-f005:**
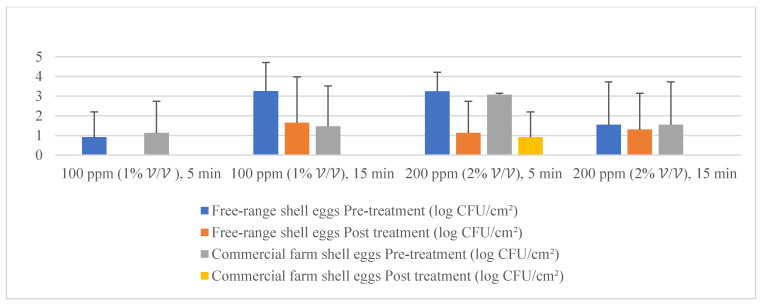
Mean population of yeasts (pre- and post-treatment) for free-range shell eggs and commercial farm shell eggs.

**Figure 6 molecules-26-05259-f006:**
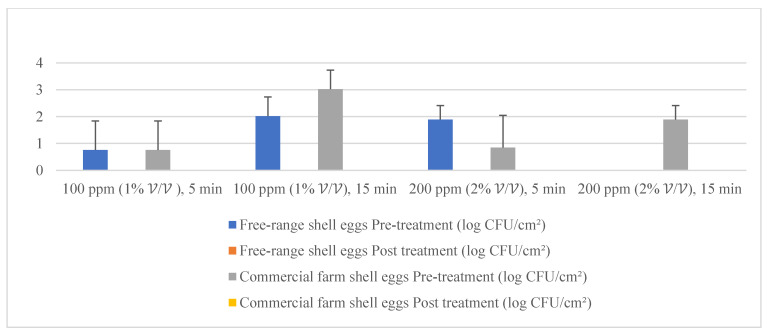
Mean population of molds (pre- and post-treatment) for free-range shell eggs and commercial farm shell eggs.

**Figure 7 molecules-26-05259-f007:**
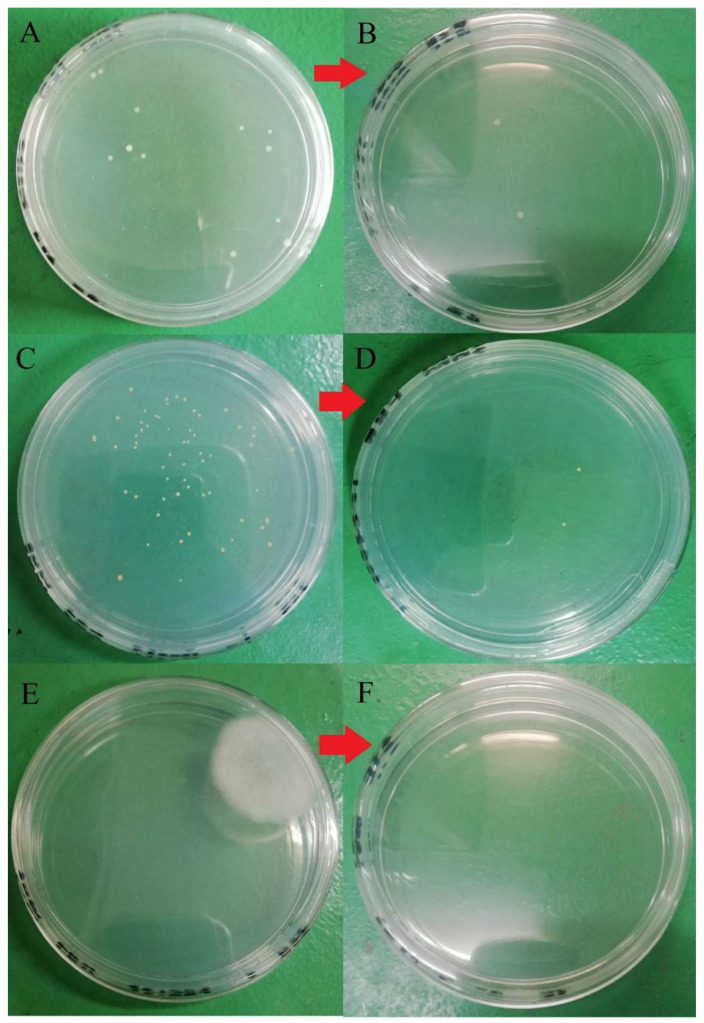
Microbial load before and after washing with 100 ppm quat sanitizer for 5 min. (**A**) Total aerobic mesophilic bacteria at a dilution of 10^5^ before washing; (**B**) total aerobic mesophilic bacteria at a dilution of 10^3^ after washing; (**C**) yeast population at a dilution of 10^1^ before washing; (**D**) yeasts population at a dilution of 10^1^ after washing; (**E**) molds population at a dilution of 10^1^ before washing; (**F**) molds population at dilution 10^1^ after washing.

**Figure 8 molecules-26-05259-f008:**
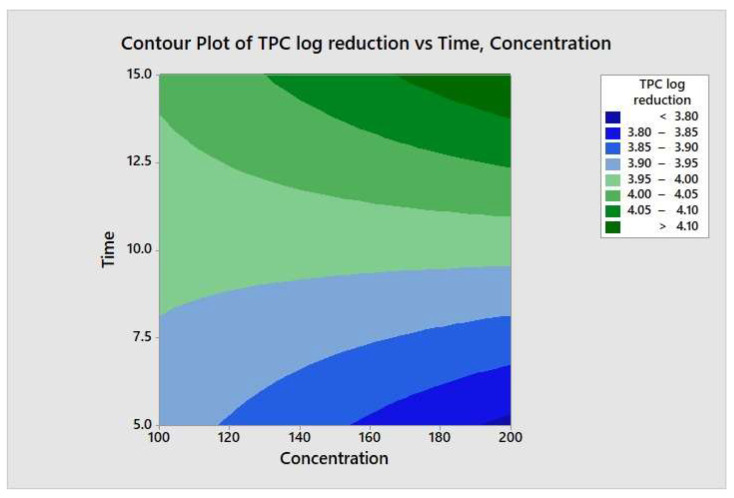
Contour plot of the total aerobic mesophilic bacteria logs reduction versus time and concentration.

**Figure 9 molecules-26-05259-f009:**
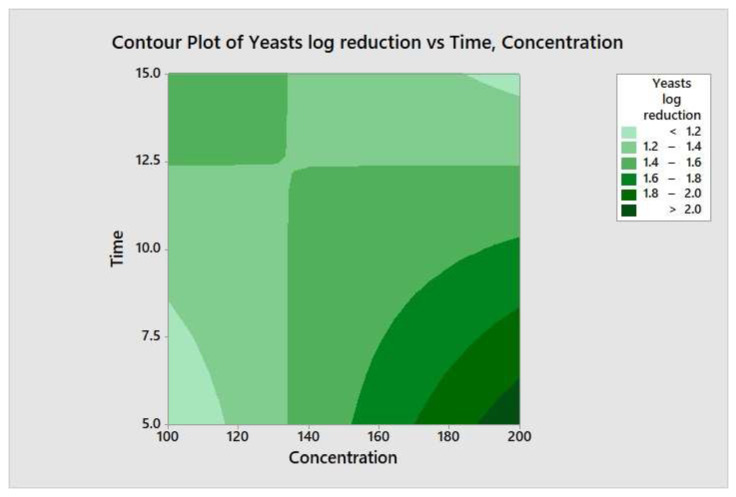
Contour plot of the yeasts logs reduction versus time and concentration.

**Figure 10 molecules-26-05259-f010:**
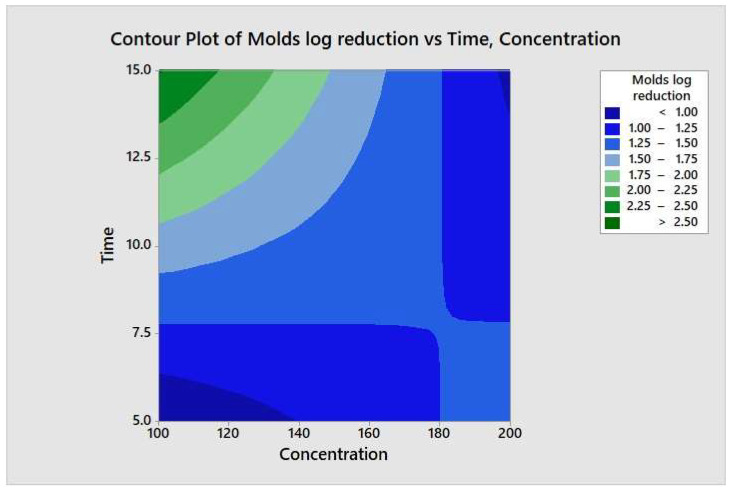
Contour plot of the molds logs reduction versus time and concentration.

**Figure 11 molecules-26-05259-f011:**
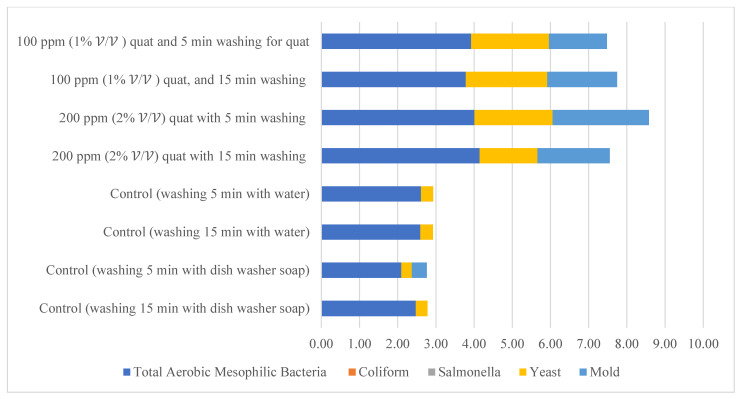
Comparison of the total reduction of microbial load (log CFU/cm^2^) after washing with quat, distilled water (control) and dishwashing soap (control).

**Figure 12 molecules-26-05259-f012:**
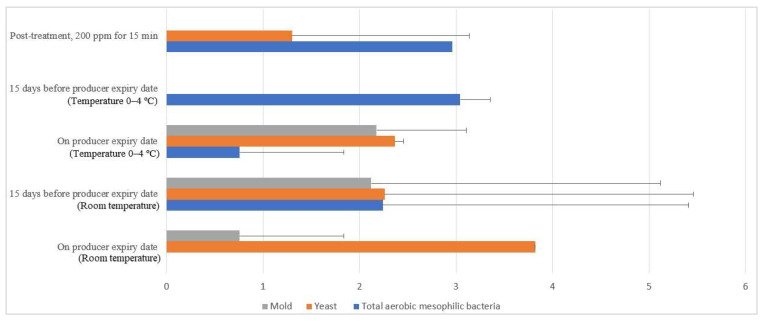
Comparison of the free-range eggshell microbial load (log10 CFU/cm^2^) during storage at room and chill temperature after washing with 200 ppm quat sanitizer for 15 min.

**Figure 13 molecules-26-05259-f013:**
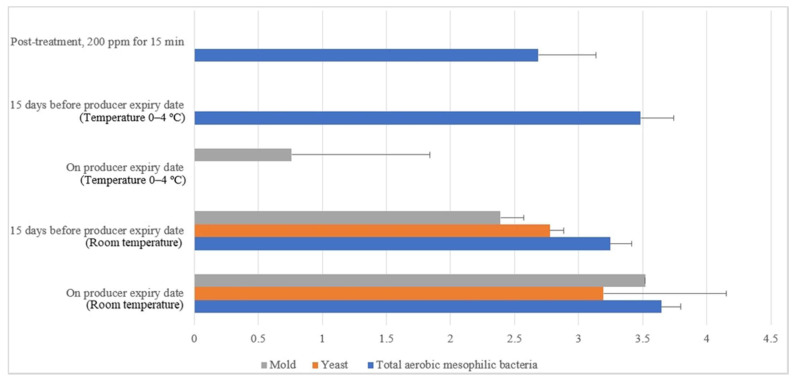
Comparison of commercial farm eggshell microbial load (log10 CFU/cm^2^) during storage at room and chill temperature after washing with 200 ppm quat sanitizer for 15 min.

**Figure 14 molecules-26-05259-f014:**
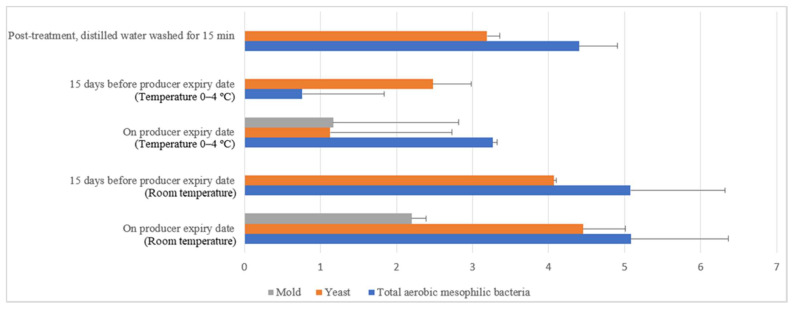
Microbial load (log10 CFU/cm^2^) during storage at room and chill temperature washed with distilled water.

**Figure 15 molecules-26-05259-f015:**
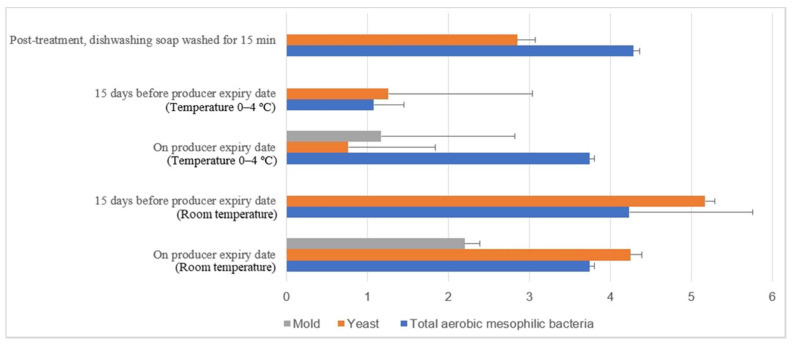
Microbial load (log10 CFU/cm^2^) during storage at room and chill temperature after washing with dishwashing soap.

**Table 1 molecules-26-05259-t001:** Samples details.

Brand	Type	Oviposition Date	Producer Expiry Date	Purchase Date	Purchase Location (GPS)
A (Producer LH)	FR ^1^	27 July 2018	25 August 2018	5 August 2018	3.117196, 101.678054
B (Producer SS)	FR	Nil ^3^	19 November 2018	14 October 2018	3.117196, 101.678054
C (Producer TS)	CF ^2^	30 July 2018	29 August 2018	8 August 2018	3.117196, 101.678054
D (Producer LH)	CF	20 August 2018	19 September 2018	21 August 2018	3.073501, 101.673531

Note: ^1^ FR = free-range, ^2^ CF = commercial farm, ^3^ Oviposition date data are not able to retrieve from the producer.

**Table 2 molecules-26-05259-t002:** Full (2^2^) factorial experiment design for the free-range egg quat sanitizer treatment.

Shell Eggs Type	Egg-Laying Date	Producer Expiry Date	*n*	Factor A (Concentration)	Factor B (Washing Time)
Brand A free-range eggs (Producer LH)	26 September 2018	26 October 2018	2	100 ppm, a_0_	5 min, b_0_
			2	100 ppm, a_0_	15 min, b_1_
			2	200 ppm, a_1_	5 min, b_0_
			8	200 ppm, a_1_	15 min, b_1_
	4 October 2018	4 November 2018	2	100 ppm, a_0_	5 min, b_0_
			2	100 ppm, a_0_	15 min, b_1_
			2	200 ppm, a_1_	5 min, b_0_
			8	200 ppm, a_1_	15 min, b_1_
Brand D commercial farm shell eggs (Producer LH)	26 September 2018	26 October 2018	2	100 ppm, a_0_	5 min, b_0_
			2	100 ppm, a_0_	15 min, b_1_
			2	200 ppm, a_1_	5 min, b_0_
			8	200 ppm, a_1_	15 min, b_1_
	4 October 2018	4 November 2018	2	100 ppm, a_0_	5 min, b_0_
			2	100 ppm, a_0_	15 min, b_1_
			2	200 ppm, a_1_	5 min, b_0_
			8	200 ppm, a_1_	15 min, b_1_

**Table 3 molecules-26-05259-t003:** Storage duration for the post-treatment microbial load study.

Shell Eggs Type	Oviposition Date	Producer Expiry Date	Microbial Growth Study on 15 Days before the Expiry Date	*n*	Microbial Growth Study on the Expiry Date	*n*
Brand A free-range egg (Producer LH)	26 September 2018	26 October 2018	11 October 2018	4 ^1^	26 October 2018	4 ^1^
	4 October 2018	4 November 2018	19 October 2018	4 ^1^	4 November 2018	4 ^1^
Brand D commercial farm shell eggs (Producer LH)	26 September 2018	26 October 2018	11 October 2018	4 ^1^	26 October 2018	4 ^1^
	4 October 2018	4 November 2018	19 October 2018	4 ^1^	4 November 2018	4 ^1^

Note ^1^ Two samples from room temperature storage and two samples from chill temperature storage.

**Table 4 molecules-26-05259-t004:** Eggshell microbial load: mean ± SD of triplicate samples.

Brand/Producer	Type	Mean Population (log10 CFU/Shell Egg)				
		Total Aerobic Mesophilic Bacteria	*Salmonella*	Coliforms	Yeasts	Molds
A/Producer LH	Free-range shell egg	10.21 ± 0.27 ^a^	ND ^1^	ND	6.96 ± 0.43 ^a^	2.00 ± 3.46 ^a^
B/Producer SS	Free-range shell egg	8.86 ± 0.25 ^b^	ND	ND	6.21 ± 0.28 ^a^	7.16 ± 0.56 ^a,b^
C/Producer TS	Commercial farm shell egg	7.94 ± 0.21 ^c^	ND	ND	6.90 ± 0.45 ^a^	4.39 ± 0.27 ^b^
D/Producer LH	Commercial farm shell egg	6.94 ± 0.49 ^d^	ND	ND	2.74 ± 0.41 ^b^	0.51 ± 0.88 ^a^

Note: ^a–d^ Values within the same column without a common superscript are significantly different (*p* < 0.05), ^1^ ND No detectable survivor cells by a direct plating procedure.

## Data Availability

Not applicable.

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
