# Peer review of "Effectiveness of Quaternary Ammonium in Reducing Microbial Load on Eggs"

_molecules, 2021, doi:10.3390/molecules26175259_

Round 1

Reviewer 1 Report

Two comments should be addressed:

  1. You are disinfecting the shell eggs with a commercial quat solution.  Why not just use dilute hypochlorite bleach?  It would be less expensive.  This needs to be explained in the Introduction.  If the same experiments have been done with bleach, this needs to be referenced, and the results with your quat studies, compared.
  2. The quat disinfection (as well as a hypochlorite disinfection) kills the pathogens on the egg shells, but once the shells dry, there is nothing to prevent new contamination during a supply chain.  A paper was published 30 years ago (J. Ind. Microbiol. 1992, 11, 37) which addressed this point with good results.  You should at least acknowledge this early paper in your Introduction and references.

Author Response

Thank you for your valuable and constructive comments. Please refer to attached documents for reply.

Reviewer 2 Report

Disinfecting food is essential. The current study indicates that washing and sanitizing shell eggs purchased from the open market is vital to minimize food poisoning incidents due to shell eggs contamination, especially preparing non-cooked/baked foods. Quat sanitizer with a maximum concentration of 200 ppm is an effective alternative to minimize the microbial load yet safe for non-rinse food contact.

This work looks like an advertisement for Quat Sanitizer Goodmaid Chemical Corp. In fact, it would be interesting to compare the effect of various disinfectants based on quaternary ammonium salts, for example, benzalkonium chloride, cetylpyridinium chloride, chlorhexidine, octenidine dihydrochloride, etc., on the microflora of the egg shell.

The work is written in an accessible language. An analysis of the effect of a quaternary ammonium solution on the microflora of eggs has been carried out. Various factors have been explored. The conclusions are drawn up correctly. The work can be published in Molecules.

Author Response

(The authors gave the same response as above.)
